# *Schistosoma haematobium* Extracellular Vesicle Proteins Confer Protection in a Heterologous Model of Schistosomiasis

**DOI:** 10.3390/vaccines8030416

**Published:** 2020-07-24

**Authors:** Gebeyaw G. Mekonnen, Bemnet A. Tedla, Darren Pickering, Luke Becker, Lei Wang, Bin Zhan, Maria Elena Bottazzi, Alex Loukas, Javier Sotillo, Mark S. Pearson

**Affiliations:** 1Centre for Molecular Therapeutics, Australian Institute of Tropical Health and Medicine, James Cook University, Cairns 4878, Queensland, Australia; gebeyaw.mekonnen@my.jcu.edu.au (G.G.M.); amarebem6@gmail.com (B.A.T.); darren.pickering@jcu.edu.au (D.P.); luke.becker@jcu.edu.au (L.B.); javier.sotillo@isciii.es (J.S.); 2Department of Medical Parasitology, School of Biomedical and Laboratory Sciences, College of Medicine and Health Sciences, University of Gondar, Gondar, Ethiopia; 3Texas Children’s Hospital Center for Vaccine Development, Department of Pediatrics and National School of Tropical Medicine, Baylor College of Medicine, Houston, TX 77030, USA; wltongji@126.com (L.W.); bzhan@bcm.edu (B.Z.); bottazzi@bcm.edu (M.E.B.); 4Parasitology Reference and Research Laboratory, Centro Nacional de Microbiología, Instituto de Salud Carlos III, Majadahonda, 28220 Madrid, Spain

**Keywords:** schistosomiasis, extracellular vesicles, tetraspanin, vaccine

## Abstract

Helminth parasites release extracellular vesicles which interact with the surrounding host tissues, mediating host–parasite communication and other fundamental processes of parasitism. As such, vesicle proteins present attractive targets for the development of novel intervention strategies to control these parasites and the diseases they cause. Herein, we describe the first proteomic analysis by LC-MS/MS of two types of extracellular vesicles (exosome-like, 120 k pellet vesicles and microvesicle-like, 15 k pellet vesicles) from adult *Schistosoma haematobium* worms. A total of 57 and 330 proteins were identified in the 120 k pellet vesicles and larger 15 k pellet vesicles, respectively, and some of the most abundant molecules included homologues of known helminth vaccine and diagnostic candidates such as *Sm*-TSP2, *Sm*23, glutathione S-transferase, saponins and aminopeptidases. Tetraspanins were highly represented in the analysis and found in both vesicle types. Vaccination of mice with recombinant versions of three of these tetraspanins induced protection in a heterologous challenge (*S. mansoni*) model of infection, resulting in significant reductions (averaged across two independent trials) in liver (47%, 38% and 41%) and intestinal (47%, 45% and 41%) egg burdens. These findings offer insight into the mechanisms by which anti-tetraspanin antibodies confer protection and highlight the potential that extracellular vesicle surface proteins offer as anti-helminth vaccines.

## 1. Introduction

Schistosomiasis is the second most important parasitic disease, only after malaria, in terms of social, economic, and public health impact [1]. *Schistosoma haematobium*, the causative agent of urogenital schistosomiasis, is highly prevalent in 53 Middle East and African countries [1] and it is also sporadically seen in India [2] and France [3]. Urogenital schistosomiasis affects more than 90 million people, mostly in sub-Saharan Africa where 180 million inhabitants are at risk, and is responsible for 150,000 deaths per year [4]. Furthermore, the most common complications associated with this disease include schistosomal hematuria, bladder wall pathology, hydronephrosis, and dysuria [4].

The current control programs against schistosomiasis are aimed at reducing the morbidity caused by the parasite by regularly treating infected populations with praziquantel [5]. Despite the efforts made to control this devastating disease, schistosomiasis is still spreading to new geographical areas [3,6]. Furthermore, praziquantel has shown reduced efficacy in field studies [7] and is not effective against the immature stages of the parasite [8,9]. Hence, a vaccine that reduces disease severity and/or reduces transmission is needed to control and eliminate schistosomiasis [10,11]. Despite efforts over decades, there is no licensed vaccine [1,12]. Even though various vaccine candidates have advanced into clinical trials targeting *Schistosoma mansoni*, the causative agent of intestinal schistosomiasis, the only vaccine candidate against *S. haematobium* to have progressed into clinical trial is a glutathione S-transferase recombinant protein, Sh28GST. However, results from a phase 3 trial conducted from 2009 to 2012 in *S. haematobium* infected children did not report any significant efficacy due to the vaccine [13]. Therefore, it is important to continue identifying new target antigens in the effort to develop a vaccine against *S. haematobium* and the other species of schistosomes [10,12].

*S. haematobium* adult worms live in the perivesicular veins where they can survive for years using evasion strategies to remain undetected by the host immune system. One of the main strategies the parasites employ to hijack the immune effector response is to release a suite of immunomodulatory excretory/secretory (ES) products. ES products comprise different proteins, glycans, lipids, and nucleic acids [14], and have been the focus of different studies aiming at understanding the molecular basis of host-parasite interactions and the subsequent use of this information to develop novel therapeutics and diagnostics [15,16,17,18]. Recently, it has been documented that the ES products from different helminths (including schistosomes) contain extracellular vesicles (EV)s [19,20,21,22]. EVs are membrane-bound organelles released by cells that can act as mediators of intercellular communication by transferring molecular signals mediated by proteins, lipids, metabolites, mRNAs, microRNAs, and other non-coding RNA species [23,24]. In addition to the transmission of information between cells within the same organism, recent studies have shown that EVs secreted by parasitic helminths are taken up by host cells within the parasite’s niche tissue and provide a means of inter-species communication [19,25,26,27,28,29,30,31,32,33,34,35,36,37,38]. For instance, EVs from trematodes and nematodes can be internalized by host cells whereupon they suppress effector immune responses [37,39,40], or in contrast, some helminth EVs contribute to pathogenesis by promoting cell proliferation and inflammatory cytokine production [29].

EVs from helminths also contain vaccine candidate antigens. For example, EVs from *S. mansoni* contain molecules that have known vaccine efficacy in animal models of schistosomiasis [27], and vaccination of mice with helminth EVs stimulates the production of protective immune responses that significantly reduce fecal egg counts, worm burdens, and symptom severity and mortality induced by infection after parasite challenge [25,41,42,43,44]. Moreover, antibodies produced against recombinant forms of *Opisthorchis viverrini* EV surface proteins hinders the uptake of EVs by cholangiocytes and suppresses the immune response that fuels pathogenesis [29,42]. 

In addition to other molecules, helminth EVs contain tetraspanins (TSPs), which have been shown to be effective vaccine candidates against *Schistosoma* spp. [21,27]. The TSPs *Sm*23, *Sm*-TSP-1, and *Sm*-TSP-2, all found in the membrane of *S. mansoni* EVs [20,27], have displayed partial efficacy in when used as adjuvanted subunit vaccines [45,46,47], and *Sm*-TSP-2 has successfully completed phase I clinical trials [48]. TSPs from other helminth species, including *S. japonicum* and *O. viverrini*, have also been shown to be efficacious vaccine candidates in animal challenge models of infection [42,49].

Although there are reports describing EVs and the vaccine efficacy of EV-derived molecules from other schistosomes, no studies have been conducted on *S. haematobium*. Herein, we have characterized for the first time the proteomic composition of small ( 120 k pellet) and large (15k pellet) subclasses of EVs from this parasite, and selected three of the EV surface TSPs for assessment as subunit vaccines in a heterologous challenge mouse challenge model of schistosomiasis.

## 2. Materials and Methods 

### 2.1. Ethics Statement

All experimental procedures performed on animals in this study were approved by the James Cook University (JCU) animal ethics committee (A2391). All experiments were performed in accordance with the 2007 Australian Code of Practice for the Care and Use of Animals for Scientific Purposes and the 2001 Queensland Animal Care and Protection Act.

### 2.2. Parasite Material and Experimental Animals

*Bulinus truncatus* snails infected with *S. haematobium* (Egyptian strain) were provided by the Biomedical Research Institute, MD, USA, and maintained in aquaria in a 27 °C incubator. Male BALB/c mice were purchased from the Animal Resource Centre, Canningvale, Western Australia, and maintained at the Australian Institute of Tropical Health and Medicine (AITHM) animal facility in cages under controlled temperature and light with free access to pelleted food and water. 

To obtain cercariae, snails were removed from the tank with a pair of forceps and washed several times with water to remove debris and rotifers, transferred to a Petri dish and incubated without water at 27 °C in the dark for 2 h. Water was then added and the snails were placed under light for 1.5 h at 28 °C. Cercariae were concentrated using a 20 µm pore size sieve and, finally, each BALB/c mouse (6 week-old) was infected with 1000 cercariae by tail penetration [50].

### 2.3. Adult Worm Culture, ES Collection, and EV Purification

*S. haematobium* adult worms were obtained by perfusion of mice at 16 weeks post-infection and parasites were washed several times with serum-free modified Basch media supplemented with 4 × antibiotic/antimycotic (SFB) [20] and then incubated in SFB (50 pairs/5 mL) at 37 °C in with 5% CO_2_ for 2 weeks. ES products were harvested daily, differentially centrifuged at 4 °C (500× *g*, 2000× *g* and 4000× *g* for 30 min each) to remove large parasite material such as eggs and tegumental debris and stored at −80 °C until use.

EVs were isolated using established methods previously described for *S. mansoni* [20]. Stored supernatants were thawed on ice, concentrated at 4 °C using a 10 kDa spin concentrator (Merck Millipore, USA) and centrifuged for 1 h at 15,000× *g* at 4 °C. The resultant pellet (containing 15 k vesicles) was washed with 1 mL of PBS, centrifuged at 15,000× *g* for 1 h at 4 °C, resuspended in 200 μL PBS and stored at −80 °C. The supernatant was ultracentrifuged at 120,000× *g* for 3 h at 4 °C using an MLS-50 rotor (Beckman Coulter, Brea, CA, USA) to collect 120 k pellet vesicles. The resultant pellet was resuspended in 70 μL of PBS and subjected to Optiprep^®^ density gradient (ODG) separation. The ODG was prepared by diluting a 60% Iodixanol solution (Optiprep^®^, Sigma-Aldrich, St. Louis, MO, USA) with 0.25 M sucrose in 10 mM Tris-HCl pH 7.2 to make 40%, 20%, 10%, and 5% iodixanol solutions, and 1.0 mL of these solutions was layered in decreasing density in an ultracentrifuge tube. The resuspended 120 k pellet vesicles were added to the top layer and ultracentrifuged at 120,000× *g* for 18 h at 4 °C. A control tube was similarly prepared using PBS instead of the 120 k pellet vesicle sample to measure the density of the different fractions recovered from the gradient. Fractions obtained from the ODG were diluted with 8 mL of PBS containing 1 × EDTA-free protease inhibitor cocktail (Santa Cruz, CA, USA), and concentrated using a 10 kDa spin concentrator to remove the excess Optiprep^®^ solution.

The density of different fractions obtained from the ODG (12 each for the sample and control) was calculated by measuring the absorbance of each fraction at 340 nm using a POLARstar Omega (BMG Labtech, Cary, NC, USA) spectrophotometer and interpolating the absorbance in a standard curve as previously shown [27]. The protein concentration of all fractions was quantified using the Quick Start™ Bradford Protein Assay Kit (Bio-Rad Laboratories, Inc., Life Science Research, Hercules, CA, USA) following the manufacturer’s instructions.

### 2.4. Determination of the Size and Concentration of EVs

The size distribution and particle concentration of the different fractions recovered after ODG, as well as the 15 k vesicle fraction, was measured using tunable resistive pulse sensing (TRPS) using a qNano instrument (Izon, Christchurch, New Zealand) following an established protocol [34]. A Nanopore NP150 and a NP400 (Izon, New Zealand) were used to measure each fraction containing 120 k pellet vesicles and the 15 k pellet vesicle fraction, respectively. Thirty-five μL of measurement electrolyte (Izon, New Zealand) was added to the upper fluid well and maximum pressure was applied; the shielding lid was clicked five to 10 times to wet the Nanopore. Then, 75 μL of measurement electrolyte was added to the lower fluid well, maximum pressure and an appropriate voltage (0.1 V) was applied, and Nanopore current was checked for stability. Thirty-five and 75 μL of filtered coating solution (Izon, New Zealand) was loaded in the upper and lower fluid well, respectively, and maximum pressure was applied for 10 min followed by maximum vacuum for another 10 min. The coating solution was flushed out of the upper and lower fluid wells two to three times with measurement electrolyte, maximum pressure was applied for 10 min and the voltage was increased until the current reached between 120 and 140 nA and the baseline current was stable. Then, 35 μL of calibration particles (CP200 carboxylated polystyrene calibration particles; Izon, New Zealand) was loaded to the upper fluid well at a 1:200 dilution when calibrating for 120 k pellet vesicle fractions and 1:1,500 when calibrating for the 15 k pellet vesicle fraction, incubated for 2 min at maximum pressure and the stretch was reduced and the calibration particles were measured at two different pressures (P10 and P5). Then, the 120 k and 15 k pellet vesicle fractions were diluted 1:5, applied to the Nanopore and measured similarly to the calibration particles. The size and concentration of particles were determined using the software provided by Izon (version 3.2).

### 2.5. In-Gel Trypsin Digestion of EVs

Fractions containing 120 k pellet vesicles of sufficient size and concentration, and the 15 k pellet vesicle fraction, were resuspended in 1 × loading buffer (10% glycerol, 80 mM Tris-HCl, 2% SDS, 0.01% bromophenol blue and 1.25% beta-mercaptoethanol, pH 6.8), boiled at 95 °C for 5 min and electrophoresed in a 15% SDS-PAGE gel at 100 V. The gel was stained with 0.03% Coomassie Brilliant Blue (40% methanol, 10% acetic acid, and 50% water) for 30 min at room temperature (RT) with gentle shaking and destained using destaining buffer 1 (60% water, 10% acetic acid, and 30% methanol) for 1 h at RT with gentle shaking. Each lane was sliced into three pieces with a surgical blade and placed into a fresh Eppendorf tube. Then, slices were further destained three times using destaining buffer 2 (50% acetonitrile (ACN), 20% ammonium bicarbonate, and 30% milliQ water) by adding 200 μL of buffer to the gel slice, and incubating them at 37 °C for 45 min. Supernatants were discarded and, finally, gel slices were dried in a vacuum concentrator (LabGear, QLD 4064, Australia) on low/high medium heat (<45 °C). Next, 100 µL of reduction buffer (20 mM dithiothreitol, 25 mM ammonium bicarbonate) was added to each dried slice, incubated at 65 °C for 1 h, and supernatants were discarded. Alkylation was achieved by adding 100 µL of alkylation buffer (50 mM iodoacetamide, 25 mM ammonium bicarbonate) to each gel slice, which were further incubated in darkness for 40 min at RT. Gel slices were washed with 200 µL of wash buffer (25 mM ammonium bicarbonate) and incubated at 37 °C for 15 min twice after the gel slices were dried in a speedivac. For trypsin digestion, a total of 2 µg of trypsin (Sigma-Aldrich, USA) was added to each gel slice and incubated for 5 min at RT. Finally, 50 µL of trypsin reaction buffer (40 mM ammonium bicarbonate, 9% ACN) was added to gel slices and incubated overnight at 37 °C. Peptides were extracted in 50% acetonitrile with 0.1% trifluoroacetic acid. The last step was performed three times to maximize peptide recovery. All peptides were finally dried in a vacuum concentrator. Samples were then resuspended in 10 μL of 0.1% trifluoroacetic acid and tryptic peptides were desalted using a Zip-Tip^®^ column (Merck Millipore, Burlington, MA, USA) pipette tip according to the manufacturer’s protocol and dried in a vacuum concentrator before analysis using liquid chromatography-tandem mass spectrometry (LC-MS/MS).

### 2.6. LC-MS/MS Analysis, Database Search, and Bioinformatic Analysis

Each 120 k pellet and 15 k pellet vesicle preparation was reconstituted in 10 μL of 5% formic acid and injected onto a 50 mm 300 μm C18 trap column (Agilent Technologies, Santa Clara, CA, USA). The samples were then desalted for 5 min at 30 μL/min using 0.1% formic acid and the peptides were then eluted onto an analytical nano-HPLC column (150 mm × 75 μm 300SBC18, 3.5 μm, Agilent Technologies, USA) at a flow rate of 300 nL/min. Peptides were separated using a 95 min gradient of 1–40% buffer B (90/10 ACN/0.1% formic acid) followed by a steeper gradient from 40 to 80% buffer B in 5 min. A 5600 ABSciex mass spectrometer operated in information-dependent acquisition mode, in which a 1 s TOF-MS scan from 350–1400 *m*/*z* was used, and for product ion ms/ms 80–1400 *m*/*z* ions observed in the TOF-MS scan exceeding a threshold of 100 counts and a charge state of +2 to +5 were set to trigger the acquisition of product ion. Analyst 1.6.1 (ABSCIEX, Framingham, MA, USA) software was used for data acquisition.

For database search and protein identification, a database was built using a concatenated target/decoy version of the *S. haematobium* predicted proteome [51,52] sequences; Bioproject PRJNA78265 downloaded from Parasite WormBase (www.parasite.wormbase.org) and concatenated to the common repository of adventitious proteins (cRAP, https://www.thegpm.org/crap/), as well as to the *Sh*-TSP-2 protein (Genbank QCO69687.1). A database search was performed using a combination of four search engines—X! Tandem version X! Tandem Vengeance (2015.12.15.2) [53], MS-GF+ version Release (v2018.04.09) [54], OMSSA, and Tide [55] using SearchGUI version v3.3.3 [56]. The identification settings were as follows: Trypsin, Specific, with a maximum of two missed cleavages 10.0 ppm as MS1 and 0.2 Da as MS2 tolerances; fixed modifications: Carbamidomethylation of C (+57.021464 Da), variable modifications: Deamidation of N (+0.984016 Da), Deamidation of Q (+0.984016 Da), Oxidation of M (+15.994915 Da). Peptides and proteins were inferred from the spectrum identification results using PeptideShaker version 1.16.38 [56]. Peptide Spectrum Matches (PSMs), peptides, and proteins were validated at a 1.0% False Discovery Rate (FDR) estimated using the decoy hit distribution. Only proteins having at least two unique peptides were considered as identified. The mass spectrometry proteomics data have been deposited in the ProteomeXchange Consortium via the PRIDE partner repository with the dataset identifier PXD019462 and 10.6019/PXD019462 (Username: reviewer46995@ebi.ac.uk Password: 03jNpGFk).

Protein family (Pfam) domains were classified using HMMER v3.1b1 [57] and protein gene ontology (GO) categories were classified using Blast2GO v5.2 [58]. ReviGO was used to visualize GO terms using semantic similarity-based scatterplots [59]. TMHMM [60] and SignalP 4.1 [61] software was used to predict transmembrane domains and putative signal peptides, respectively.

### 2.7. Cloning and Expression of Sh-TSP2 and MS3_09198 in Pichia Pastoris

DNA sequences encoding the large extracellular loop (LEL) regions of *Sh*-TSP2 and MS3_09198 (predicted using TMPRED) were codon optimized based on yeast codon usage preference and synthesized by GenScript (Piscataway, NJ, USA). The synthesized coding DNAs with a 6-His-tag expressed at the C-terminus were cloned into the pPinkα-HC expression vector (Thermofisher, Waltham, MA, USA) using *Xho*I/*Kpn*I restriction sites. The recombinant plasmids with correct insert confirmed by double-strand DNA sequencing were linearized with *Af*lII, and then transformed by electroporation into PichiaPink™ Strain 4 with endogenous proteinases A and B knocked out to prevent degradation of expressed proteins. The transformants were selected on PAD (Pichia Adenine Dropout) agar plates and the expression yield of picked colonies was evaluated in 10 mL BMMY medium with 0.5% methanol for 72 h. The clones with the highest expression yield were used to express recombinant *Sh*-TSP2 and MS3_09198 in 2L BMMY under induction of 0.5% methanol at 30 °C with 250 rpm shaking for 72 h. The culture medium containing the secreted proteins was harvested by centrifugation (5000× *g* for 20 min at RT) and filtered through a 0.22 µM membrane filter (Millipore). Recombinant proteins were purified by immobilized metal affinity chromatography (IMAC) using a prepacked 5 mL His-Trap HP column (GE Healthcare, Chicago, IL, USA). The purified recombinant proteins were buffer-exchanged in 1 × PBS, pH 7.4. The purity of the proteins was analyzed by SDS-PAGE and the concentration was measured by OD_280_. The purified recombinant proteins were adjusted to 1 mg/mL, aliquoted and stored at −80 °C. 

### 2.8. Cloning and Expression of MS3_01370 in Escherichia Coli

Unlike *Sh*-TSP2 and MS3_09198, we were unsuccessful at expressing the LEL region of MS3_01370 in soluble form in *P. pastoris*, thus, the protein was expressed as a thioredoxin (TrX) fusion in *E. coli*. Primers incorporating *Nco*I (forward primer) and *Xho*I restriction enzyme sites (reverse primer) were used to amplify the LEL of MS3_01370 from *S. haematobium* cDNA and the amplicon was cloned into the pET32a expression vector (Novagen), in-frame with the N-terminal TrX tag. Protein expression was induced for 24 h in *E. coli* BL21(DE3) by the addition of 1 mM Isopropyl beta-D-1-thiogalactopyranoside (IPTG) using standard methods. The culture was harvested by centrifugation (8000× *g* for 20 min at 4 °C), re-suspended in 50 mL lysis buffer (50 mM sodium phosphate, pH 8.0, 300 mM NaCl, 40 mM imidazole) and stored at −80 °C. The cell pellet was lysed by three freeze-thaw cycles at −80 °C and 42 °C followed by sonication on ice (10 × 5 sec pulses [70% amplitude] with 30 s rest periods between each pulse) with a Qsonica Sonicator. Insoluble material was pelleted by centrifugation at 20,000× *g* for 20 min at 4 °C. The supernatant was diluted 1:4 in lysis buffer and filtered through a 0.22 μm membrane (Millipore). MS3_01370 was purified by IMAC by loading onto a prepacked 1 mL His-Trap HP column (GE Healthcare) equilibrated with lysis buffer at a flow rate of 1 mL/min using an AKTA-pure-25 FPLC (GE Healthcare). After washing with 20 mL lysis buffer, bound His-tagged protein was eluted using the same buffer with a stepwise gradient of 50–250 mM imidazole (50 mM steps). Fractions containing MS3_01370 (as determined by SDS-PAGE) were pooled and concentrated using Amicon Ultra-15 centrifugal devices with a 3 kDa MWCO and quantified using the Pierce BCA Protein Assay kit. The final concentration of MS3_01370 was adjusted to 1 mg/mL and the protein was aliquoted and stored at −80 °C.

### 2.9. Vaccine Formulation and Immunization Schedule

Numbers of *S. haematobium* cercariae were insufficient to perform appropriately powered vaccination trials using an *S. haematobium* model of infection so vaccine experiments were instead performed using an *S. mansoni* challenge model [62]. Four groups of 10 BALB/c mice (6–8 weeks) were immunized intraperitoneally on day 1 with either recombinant *Sh*-TSP-2, MS3_09198, MS3_01370, or TrX control protein (50 μg/mouse), each formulated with an equal volume of Imject alum adjuvant (Thermofisher) and 5 μg of CpG ODN1826 (InvivoGen, San Diego, CA, USA). Immunizations were repeated on days 15 and 29 and each mouse was challenged (tail penetration) with 120 *S. mansoni* cercariae on day 43. Blood was sampled at day 42 to determine pre-challenge antibody titers. Two independent trials were performed.

### 2.10. Necropsy and Estimation of Parasite Burden

Mice were necropsied at day 91 (7 weeks post-infection) [62]. Blood was collected and worms harvested by vascular perfusion and counted. Livers were removed, weighed, and digested for 5 h with 5% KOH at 37 °C with shaking. Schistosome eggs from digested livers were concentrated by centrifugation at 1000× *g* for 10 min and re-suspended in 1 mL of 10% formalin. The number of eggs in a 5 μL aliquot was counted in triplicate and the number of eggs per gram (EPG) of liver tissue was calculated. Small intestines were removed and cleaned of debris before being weighed and digested as per the livers. Eggs were similarly concentrated and counted to calculate intestinal EPG.

### 2.11. Statistics

All statistics were performed using GraphPad Prism 7.0. The reductions in worm and egg numbers were analyzed using a Student’s *t*-test and results were expressed as the mean ± standard error of the mean. For antibody titers, the reactivity cut-off values were determined as the mean + 3SD of the naive serum.

## 3. Results

### 3.1. Density, Protein Concentration, Particle Concentration, and Purity of 120 k and 15 k Pellet Vesicles from Schistosoma haematobium

*S. haematobium* adult worm 120 k pellet vesicles were purified using an iodixanol gradient. The density of the 12 fractions obtained after gradient separation ranged from 1.039 to 1.4 g/mL (Appendix A). The protein and particle concentration of the fractions ranged from 1.6 to 25.35 μg/mL and 3.72 × 10^6^ to 2.06 × 10^8^ particles/mL, respectively, while the protein and particle concentration of the 15 k pellet vesicle fraction was 18.00 μg/mL and 1.48 × 10^7^ particles/mL, respectively. The size of the particles in the gradient-separated fractions ranged from 135 nm ± 19.3 to 342 nm ± 113.9 and size of particles in the 15 k pellet vesicle fraction was 274 nm ± 40.7. Gradient-separated fractions having an appropriate purity and density corresponding to 120 k pellet vesicles (1.09–1.22 g/mL) [20,27] (fractions 5–9) were selected for further analysis (Figure 1). 

### 3.2. Proteomic Analysis of Schistosoma haematobium 120 k and 15 k Pellet Vesicles

The proteome composition of *S. haematobium* adult worm EVs was characterized by LC-MS/MS. After combining the protein identified in fractions 5–9 (fractions containing the highest purity of 120 k pellet vesicles), a total of 133 proteins matching *S. haematobium* proteins and common contaminants from the cRAP database were identified. From these, 80 proteins were identified with at least two validated unique peptides and 57 of them matched *S. haematobium* proteins. From the 57 identified proteins, eight (14%) contained a transmembrane domain and seven (12%) had a signal peptide. In a similar fashion, 506 proteins were identified from analysis of the 15 k pellet vesicles. From these, 344 proteins were identified with at least two validated unique peptides and 330 matched *S. haematobium* proteins. From these identified proteins, 54 (16.3%) contained a transmembrane domain and 30 (9%) had a signal peptide. Forty proteins were identified in both types of vesicles. The identity of the most abundant proteins in each type of vesicle, as well as proteins having homologues typically found in other helminth EVs, are shown in Table 1. A full list of proteins identified in both 15 k and 120 k pellet vesicles is shown in Appendix A.

### 3.3. Protein Families Present in Schistosoma haematobium 120 k and 15 k Pellet Vesicles

Identified proteins were subjected to a Pfam analysis using default parameters in HMMER v3.1b1 and proteins containing an identified Pfam domain with an *E*-value < 1 × 10^−5^ were selected. A total of 70 and 387 domains were identified from 120 k and 15 k pellet vesicles, respectively. In 120 k pellet vesicles, the three most abundant domains were proteasome subunit domains (PF00227) (14%), TSP family domains (PF00335) (7%), and ferritin-like domains (PF12902) (4%) (Figure 2A). The most abundant protein domains from 15 k pellet vesicles were EF-hand domains (PF00036) (3%), Ras family domains (PF00071) (3%), TCP-1/cpn60 chaperonin family domains (PF00118) (2%), and TSP family domains (PF00335) (2%) (Figure 2B). From these, TSP family domains and 14-3-3 protein domains were common to both types of vesicles.

### 3.4. Gene Ontology of Proteins Identified from Schistosoma haematobium 120 k and 15 k Pellet Vesicles

The proteins of adult *S. haematobium* 120 k and 15 k pellet vesicles were annotated using Blast2GO. To avoid redundancy in the analysis and better comprehend the represented GO terms in the vesicles, the parental GO terms were removed and children GO terms were visualized using ReviGO based on semantic similarity-based scatterplots. The GO terms were ranked by the nodescore provided by Blast2GO and plotted using their nodescore and frequency. Semantically similar GO terms plot close together and increasing heatmap score signifies increasing nodescore from Blast2GO. The circle size denotes the frequency of the GO term from the underlying database. In 120 k pellet vesicles, several biological processes were highly represented, such as the ubiquitin-dependent protein catabolic process, oxidation-reduction process, and gluconeogenesis and glycolytic process (Figure 3A). Similarly, in 15 k pellet vesicles, several biological processes were highly represented, such as the carbohydrate metabolic process, transport process, organonitrogen compound metabolic process, and microtubule-based process (Figure 3B). The oxidation–reduction process was common to both types of vesicles; six proteins from 120 k pellet vesicles and 22 proteins from 15 k pellet vesicles were predicted to be involved in this process.

In *S. haematobium* 120 k vesicles, several molecular functions were highly represented, such as threonine-type endopeptidase activity, protein binding activity, endopeptidase activity, and transition metal ion binding activity. In 15 k vesicles, molecular functions such as protein binding activity, ATP binding activity, nucleoside-triphosphatase activity, and calcium ion binding activity were highly represented. From these highly represented molecular function terms, protein binding was common to both 120 k and 15 k vesicles.

### 3.5. Parasite Burdens in Vaccinated and Control Mice

Vaccination of mice with MS3_01370, MS3_09198 showed a trend towards reduced adult *S. mansoni* burden in trial 1 by 22% and 12%, respectively, and in trial 2 by 14% and 5%, respectively, compared to controls. *Sh*-TSP2-vaccinated mice showed a reduction in trial 1 only of 2%, compared to controls. None of these differences were statistically significant except for MS3_01370 in trial 1 (*p* < 0.05) (Figure 4A,D). However, in trial 1, vaccination of mice with MS3_01370, MS3_09198 and *Sh*-TSP-2 significantly reduced liver egg burdens by 39% (*p* < 0.05), 49% (*p* < 0.001), and 32% (*p* < 0.01), respectively (Figure 4B), and in trial 2, MS3_01370, MS3_09198, and *Sh*-TSP-2 vaccination significantly reduced liver egg burdens by 54% (*p* < 0.001), 27% (*p* < 0.05), and 49% (*p* < 0.001), respectively (Figure 4E). Similarly, immunization of mice with MS3_01370, MS3_09198 and *Sh*-TSP-2 reduced the intestinal egg burden by 57% (*p* < 0.01), 51% (*p* < 0.01), and 54% (*p* < 0.001) in trial 1, respectively (Figure 4C), and in trial 2, MS3_01370, MS3_09198, and *Sh*-TSP-2 vaccination reduced the intestinal egg burden by 36% (*p* < 0.01), 39% (*p* < 0.05), and 27% (*p* < 0.05), respectively (Figure 4F).

## 4. Discussion

More than 100 million people are infected with *S. haematobium* and 150,000 people die every year [4]. Efforts have been made to reduce the prevalence of schistosomiasis, but the parasite is spreading to new areas [3] and is now the second most prevalent of the neglected tropical diseases [63]. Praziquantel is the only drug available for schistosomiasis treatment, although it does not protect against re-infection and the risk of resistance emerging in the field is high [64,65]. Furthermore, there is no licensed and effective vaccine for this devastating disease, and there is an urgent need to develop one to eliminate urogenital schistosomiasis as well as the other species of schistosomes.

We have shown previously that *S. haematobium* secretes at least two populations of vesicles with distinct differences in size and proteomic composition, similar to other schistosome species such as *S. mansoni* [20]. In this paper, we now show the first proteomic analysis of two types of extracellular vesicles (exosome-like, 120 k pellet vesicles and microvesicle-like, 15 k pellet vesicles) from adult *Schistosoma haematobium* worms. 

The most represented domains contained within 120 k pellet vesicle proteins were the proteasome subunit domains, TSP domains, and ferritin-like domains, among others, while the most represented domains in the 15 k pellet vesicles were EF-hand, Ras family, TCP-1/cpn60 chaperonin family and TSP family domains. The proteasome is involved in the biogenesis of EVs [66] and also controls protein homeostasis and degradation of damaged proteins [67]. Furthermore, in schistosomes, the proteasome plays an important role in the cellular stress response and survival of the parasite [68]; it has been shown that treating mice with a proteasome inhibitor prior to infection with *S. mansoni* cercariae significantly impaired parasite development [69] and in vitro treatment of schistosomula with siRNAs targeting a deubiquitinase subunit of the 19S regulatory particle significantly reduced parasite viability [70]. 

Ferritins are iron-storage proteins, involved in maintaining intracellular iron balance [71], which minimize free-radical reactions and prevent cellular damage caused by iron accumulation in the cell [72]. Iron also plays an important role in the eggshell formation of schistosomes, and fer-1, one of the two ferritin isoforms of the parasite, is highly expressed in female worms in comparison to males [73]. Since female worms produce many eggs per day and eggs are the primary cause of pathology, vaccination using ferritins could disrupt the formation of eggs and reduce egg-induced disease. Indeed, ferritins have been tested as vaccine candidates against schistosomes; immunization of mice with the recombinant Fer-1 of *S. japonicum* caused 35.5% and 52.1% reduction in adult worm and liver egg burden, respectively [74]. Ferritins have been identified in the proteomic analysis of other blood feeding helminth EVs [20,27], suggesting a role for EVs in iron storage and acquisition. 

Proteins containing EF-hand domains are involved in a number of protein-protein interactions for the uptake and release of calcium [75]. The influx of calcium in the cell induces the redistribution of phospholipids in the cell membrane, resulting in increased release of microvesicles [76]. EF-hand domains are also among the most predominant protein domains found within other helminth EV proteins [77]. This is consistent with the GO analysis herein, in which proteins involved in protein binding and calcium ion binding are the highest represented molecular function terms.

Ras proteins serve as signaling nodes activated in response to diverse extracellular stimuli [78] and are involved in biogenesis and release of microvesicles [79]. In *S. mansoni*, Ras proteins are involved in the male-directed maturation of the female worms [80], which could suggest a potential role of EVs in parasite-parasite communication.

TCP-1/cpn60 chaperonin family proteins play an important role in the folding of proteins, including actin and tubulin [81], which bind and hydrolyze ATP using magnesium ions [82]. This is consistent with the molecular function GO terms, in which ATP binding and nucleoside-triphosphatase was the most represented.

The second and fourth most abundant protein domains in 120 k and 15 k pellet vesicles, respectively, were the TSPs. TSPs are involved in EV biogenesis [79], are present on the surface membrane of EVs from many different organisms, and are considered a molecular marker of EVs [83]. TSPs are also found from the proteomic analysis of other helminth EVs (reviewed in [77]). In trematodes, TSPs are involved in tegument development [84,85,86]. TSP LELs have been tested as vaccine candidates in trematode models of infection [42,47,87] and antibodies produced against TSP vaccine candidates present in *S. mansoni* and *O. viverrini* EVs blocked the internalization of EVs through host cells in both parasites, and decreased pathogenesis in the case of *O. viverrini* [29,42,88], suggesting a possible mechanism of vaccine efficacy. 

Tetraspanins were highly represented in the analysis and found in both vesicle types. Since there is evidence that these molecules orchestrate interactions between parasite EVs and host tissues, we aimed to assess the vaccine efficacy of some of the most abundant EV TSPs in a challenge model of schistosome infection. Difficulties in generating the large amount of cercariae needed for mouse infection (the use of hamsters is prohibited in Australia) precluded the use of *S. haematobium* as the challenge parasite, and thus, we used an *S. mansoni* challenge model instead. Although this meant that a heterologous infection model would be employed to assess the efficacy of *S. haematobium* proteins, we reasoned that the homology of the candidates between the two species could be sufficiently high (*Sh*-TSP2—69.6% identity, MS3_09198—98.6% identity, and MS3_01370—79.5% identity) (Appendix A) as to afford a level of cross-species protection and provide informative data with regards to the vaccine efficacy of these candidates.

Vaccination with any of the TSPs did not elicit significant reductions in worm burden compared to controls (except for MS3_01370 in trial 1); however, significant reductions in tissue egg burdens (both liver and intestinal) were observed in all groups of vaccinated mice in both trials. Decreases in tissue egg loads are arguably the most important hallmarks of an effective vaccine against schistosomiasis given that (1) pathology due to the disease is egg-induced [89] and (2) disease transmission is dependent on the excretion of eggs from the host into the environment [10], thus, a vaccine that reduces egg burden in the host would ameliorate both disease pathology and transmission.

Other schistosome antigens have been reported to elicit a primarily anti-fecundity protective effect upon vaccination. Glutathione-S-transferases (GSTs) from *S. japonicum*, *S. haematobium* and the bovine schistosome *S. bovis* have all been reported to induce decreases in tissue egg loads, despite no observable reduction in worm burden [90,91,92]. Interestingly, GSTs have been prominently identified from the vesicles in this study and other proteomic analyses of schistosome EVs [20,27,28]. This manifestation of protective immunity is seen in cattle that have had repeated field and laboratory exposure to *S. bovis* [93,94] and the closely-related *S. matheei* [95], in Rhesus monkeys experimentally infected with *S. mansoni* [96], and in baboons following laboratory exposure to *S. haematobium* [97]. In the cattle/*S. bovis* and baboon/*S. haematobium* models, this observation was confirmed by surgically transplanting “suppressed” worms into hosts with no prior exposure and showing that the parasites resumed high levels of egg production [91,97]. It could be that vaccination with the EV molecules described herein is eliciting similar immune mechanisms to engender the observed anti-fecundity effects, which may be manifesting in a reduction of the fitness and fecundity of adult worms and/or a shortening of the lifespan of eggs embolized in the tissues [91]. Furthermore, we have reported on the capacity of helminth EVs to promote pathogenesis via their endocytosis by host cells and subsequent manipulation of host immune effector mechanisms, such as an increase in IL-6 secretion [29] and differential expression of genes involved in immune cell trafficking [88], and the ability of antibodies against helminth EV proteins, such as TSPs, to block vesicle uptake [29,42,88] provides a plausible mechanism for protective efficacy.

In this study we provide the first characterization of EVs secreted by *S. haematobium*, their expression as vaccine candidates present on the surface of these EVs, and the evaluation of these candidates in a heterologous challenge model of schistosomiasis vaccination of mice with recombinant versions of three of these tetraspanins-induced protection in a heterologous challenge (*S. mansoni*) model of infection, resulting in significant reductions (averaged across two independent trials) in liver (47%, 38%, and 41%) and intestinal (47%, 45%, and 41%) egg burdens. The significant reduction in tissue egg burden described here indicates that these EV-derived vaccine candidates could be effective in reducing the pathology and transmission of *S. mansoni* and *S. haematobium* (due to the cross-protective efficacy observed in the heterologous vaccine/challenge model herein) and they could potentially be incorporated into a pan-schistosome vaccine, due to the geographical overlap between the two species.

Currently, the World Health Organization (WHO) roadmap regarding schistosomiasis is to eliminate disease morbidity and mortality as a public health problem in the coming decade [98]. Vaccine-linked chemotherapy has been deemed one of the most effective control measures for combating the disease, with this strategy combining the mainstay of schistosomiasis intervention (mass administration of PZQ) with the more long-term benefits of vaccination, such as the one based on the molecules described herein. Modeling has predicted that chemotherapy linked with a vaccine that has an effective duration of protection (transmission interruption and pathology reduction) is the best intervention strategy for achievement of the WHO’s goals to significantly decrease the burden of disease caused by schistosomiasis [99].

## Figures and Tables

**Figure 1 vaccines-08-00416-f001:**
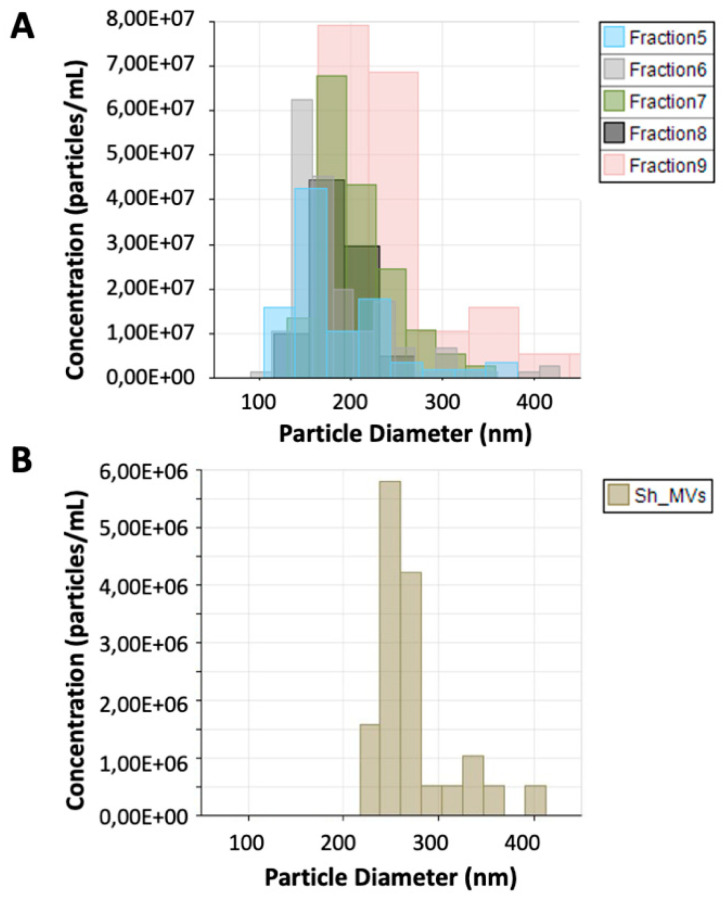
Tunable resistive pulse sensing analysis of 120 k pellet vesicles and 15 k pellet vesicles from *Schistosoma haematobium*. The size and number of extracellular vesicles (EVs) secreted by *S. haematobium* were analyzed by qNano (iZon). (**A**) Size and concentration of particles in fractions containing 120 k pellet vesicles (5–9). (**B**) Size and particle concentration of *S. haematobium* 15 k pellet vesicles.

**Figure 2 vaccines-08-00416-f002:**
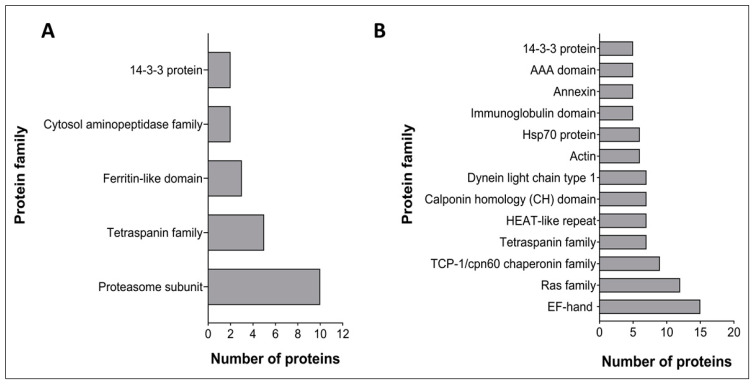
Pfam analysis of the most abundant *Schistosoma haematobium* vesicle proteins. The X-axis represents the number of proteins containing at least one of those domains. (**A**) 120 k pellet vesicles (**B**) 15 k pellet vesicles.

**Figure 3 vaccines-08-00416-f003:**
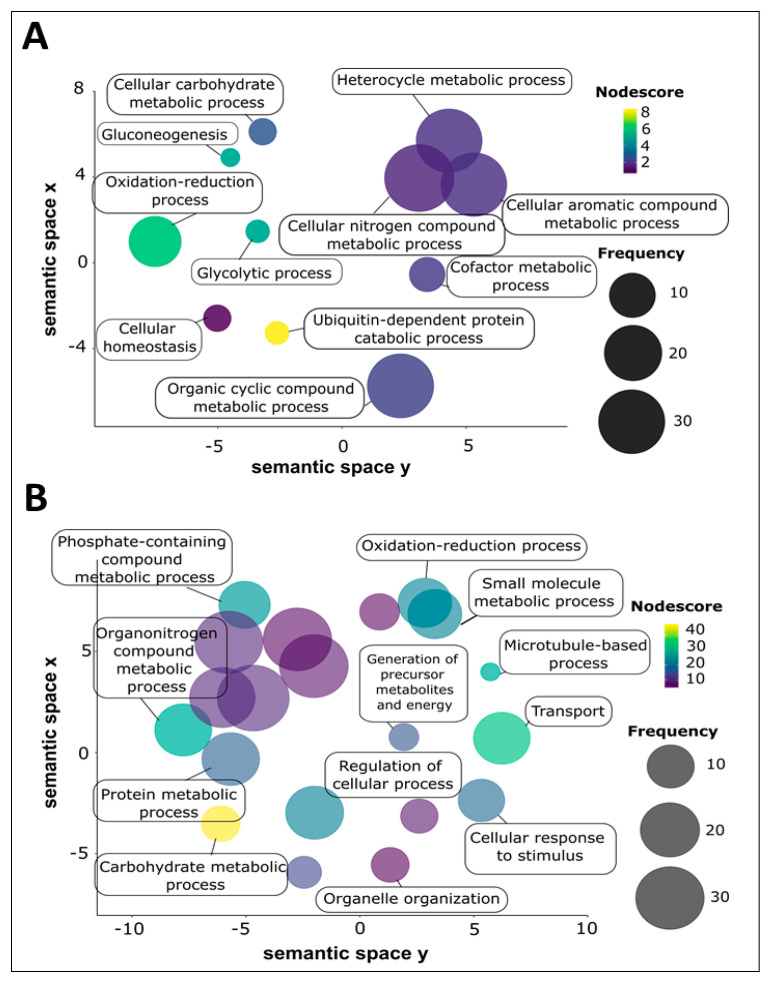
Biological process GO term categories of adult *Schistosoma haematobium* vesicle proteins. Biological processes were ranked by nodescore (Blast2GO) and plotted using REViGO. Semantically similar GO terms plot close together, increasing heatmap score signifies increasing nodescore from Blast2GO, while circle size denotes the frequency of the GO term from the underlying database. (**A**) 120 k pellet vesicles, (**B**) 15 k pellet vesicles.

**Figure 4 vaccines-08-00416-f004:**
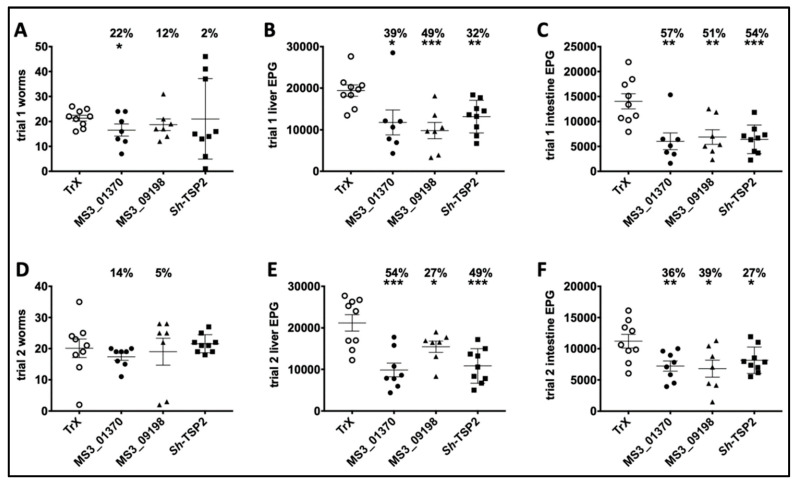
*Schistosoma mansoni* worm and egg burden reduction of vaccinated and control mice vaccinated with *S. haematobium* recombinant tetraspanins. (**A**) Adult worm reduction trial 1, (**B**) liver egg reduction trial 1, (**C**) intestinal egg reduction trial 1, (**D**) Adult worm reduction trial 2, (**E**) liver egg reduction trial 2, and (**F**) intestinal egg reduction trial 2. The percentage of reductions in parasite burden are above each dataset. Differences between each vaccinated group and the control group were analyzed with a student’s t-test. * *p* < 0.05, ** *p* < 0.01, *** *p* < 0.001.

**Table 1 vaccines-08-00416-t001:** Proteins found in 120 k and 15 k pellet vesicles isolated from the ES products of adult *S. haematobium* *.

Protein Category	Protein Accession Numbers
** 120 k vesicles**	
Proteasome subunit	MS3_10249.1, MS3_05734.1, MS3_01483.1, MS3_06009.1, MS3_04526.1, MS3_08808.1, MS3_07240.1, MS3_02807.1, MS3_09236.1, MS3_03070.1
GAPDH	MS3_10141.1
Papain family cysteine protease	MS3_08498.1
C-terminal domain of 1-Cys peroxiredoxin	MS3_08460.1
Ferritin-like domain	MS3_08059.1
S-adenosyl-L-homocysteine hydrolase	MS3_04449.1
Cytosol amino peptidase	MS3_01749.1
Trefoil (P-type) domain-containing protein	MS3_00004.1
**15k vesicles**	
EF hand	MS3_05735.1, MS3_00180.1, MS3_09846.1, MS3_05877.1, MS3_05317.1, MS3_04536.1, MS3_10043.1, MS3_05959.1, MS3_05150.1, MS3_04275.1, MS3_05958.1 MS3_05952.1, MS3_00361.1, MS3_02003.1
Ras family	MS3_10193.1, MS3_05953.1, MS3_05910.1, MS3_05976.1, MS3_07854.1, MS3_11139.1, MS3_02375.1, MS3_01653.1, MS3_04355.1, MS3_09110.1, MS3_09593.1, MS3_03443.1
TCP-1/cpn60 chaperonin family	MS3_03054.1, MS3_06928.1, MS3_01627.1, MS3_10572.1, MS3_06669.1, MS3_07556.1, MS3_08399.1, MS3_00785.1, MS3_08926.1
Tetraspanins	MS3_01905.1, MS3_01370
Heat-like repeat	MS3_08696.1, MS3_01642.1, MS3_09658.1, MS3_10590.1, MS3_05814.1, MS3_02928.1, MS3_06293.1
Calponin homology (CH) domain	MS3_07481.1, MS3_05505.1, MS3_01744.1, MS3_00852.1, MS3_00361.1, MS3_03766.1, MS3_10701.1
Dynein light chain type 1	MS3_05351.1, MS3_08569.1, MS3_05345.1, MS3_01173.1, MS3_05342.1, MS3_04412.1, MS3_05960.1
Actin	MS3_07374.1, MS3_04014.1, MS3_00351.1, MS3_02465.1, MS3_04907.1, MS3_01922.1
HSP-70 protein	MS3_10713.1, MS3_11293.1, MS3_11411.1, MS3_10049.1, MS3_02688.1, MS3_02787.1
Immunoglobulin domain	MS3_03027.1, MS3_01271.1, MS3_03208.1, MS3_07594.1, MS3_01223.1
Annexin	MS3_08725.1, MS3_08723.1, MS3_04598.1, MS3_01964.1, MS3_01952.1
AAA domain	MS3_03802.1, MS3_02581.1, MS3_01139.1, MS3_01650.1, MS3_07031.1
14-3-3 protein	MS3_03977.1, MS3_05219.1, MS3_00047.1, MS3_01871.1, MS3_03976.1
** 120 k and 15 k vesicles**	
Tetraspanins	MS3_09198, *Sh*-TSP-2, MS3_05226, MS3_05289, MS3_01153
Ferritin-like domain	MS3_07972.1, MS3_07178.1
14-3-3 protein	MS3_03977.1, MS3_00047.1
Elongation factor Tu C-terminal domain	MS3_08479.1
EF hand	MS3_08446.1
Actin	MS3_07374.1
GST, N-terminal domain	MS3_06482.1
Cytosol aminopeptidase family, catalytic domain	MS3_08450.1
Lipocalin/cytosolic fatty-acid binding protein family	MS3_04307.1
Immunoglobulin domain	MS3_03208.1
Saposin-like type B, region 2	MS3_02805.1
Enolase, N-terminal domain	MS3_02425.1

* proteins listed are members of protein families typically found in helminth EVs.

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
