# Peer review of "Schistosoma haematobium Extracellular Vesicle Proteins Confer Protection in a Heterologous Model of Schistosomiasis"

_vaccines, 2020, doi:10.3390/vaccines8030416_

Round 1

Reviewer 1 Report

Schistosomiasis is now the second most prevalent of the neglected tropical diseases, so the topic of the present study is significant. The authors present some important and novel findings regarding the proteomic analysis by LC-MS/MS of EVs derived from Schistosoma haematobium worms and potential use of tetraspanins commonly expressed in these EVs for developing a new vaccination against helminth parasites. However, there are some concerns with the conclusions in this manuscript, which need to be addressed.    1. In Fig. 1, S. haematobium adult worm 120k pellet EVs, termed exosome-like EVs, were analyzed by density gradient separation. However, 15k pellet EVs were not analyzed in the same manner. It seems MV-like EVs carry more proteins, but why the 15k pellet fraction was not analyzed by density gradient separation is unclear. Is this EV fraction more uniform compared to 120k pellet EVs?    2. The authors provided only protein concentrations and particle sizes in each fraction after the density-dependent separation. Is it possible to analyze the distribution of EV-associated proteins in these fractions by western blotting to analyze the heterogeneity of 120k pellet EVs? Besides, the authors called the MV-like subpopulations as larger 15k pellet vesicles, but the size in Fig. 1B shows that the majority of 15k pellet EVs are smaller than 120k pellet EVs in Fig. 1A. This point needs to be clarified.   3. The authors described that TSP family domains, ferritin-like domains, and 14-3-3 protein domains were common to both types of vesicles. However, the ferritin-like domain was not listed in Fig. 2B.   4. As the authors described, tetraspanin family is common to both EV fractions, but their abundance is not the highest among the listed proteins. The rationale of why tetraspanins were chosen as potential vaccine candidates should be described.    5. Why is the animal study in Fig. 4 shown as two separate trials? The rationale for this duplication should be explained. Moreover, this result could be an important finding but the underlying mechanism of this protection was not studied at all in this study. Studying the protection mechanism by injecting EV-derived tetraspanins would add significance to the study - for instance, immune cell activation by tetraspanins in mice. 

Author Response

  1. In Fig. 1, S. haematobium adult worm 120k pellet EVs, termed exosome-like EVs, were analyzed by density gradient separation. However, 15k pellet EVs were not analyzed in the same manner. It seems MV-like EVs carry more proteins, but why the 15k pellet fraction was not analyzed by density gradient separation is unclear. Is this EV fraction more uniform compared to 120k pellet EVs?

The scientific field now agrees that at least two different methods should be combined to isolate exosome-like EVs to obtain a highly pure fraction that could be used in subsequent studies (see the 2018 MISEV guidelines https://www.tandfonline.com/doi/full/10.1080/20013078.2018.1535750). Therefore, we combined ultracentrifugation and Optiprep gradient centrifugation for the separation of exosome-like vesicles from larger microvesicles and protein aggregates having comparable sedimentation velocities that can pellet at 120,000 g. The low speed used to pellet microvesicle-like EVs (termed 15k EVs in our study) does not usually allow co-precipitation of small complexes of proteins, and the combination of differential centrifugation at 500 g, 2,000 g and 4,000 g for 30 min each allows the removal of bigger material that could potentially co-precipitate at 15,000 g. The 15k EV population has been described in other trematodes such as Fasciola hepatica (https://www.ncbi.nlm.nih.gov/pmc/articles/PMC4762619/) and even in the related species Schistosoma mansoni (https://www.sciencedirect.com/science/article/pii/S0166685119301574), and we have followed the same approaches for the isolation of this population of EVs.

  1. The authors provided only protein concentrations and particle sizes in each fraction after the density-dependent separation. Is it possible to analyze the distribution of EV-associated proteins in these fractions by western blotting to analyze the heterogeneity of 120k pellet EVs? Besides, the authors called the MV-like subpopulations as larger 15k pellet vesicles, but the size in Fig. 1B shows that the majority of 15k pellet EVs are smaller than 120k pellet EVs in Fig. 1A. This point needs to be clarified.

Our data shows that the density of the fractions having the highest purity do match the ones observed for the related species S. mansoni

(https://www.sciencedirect.com/science/article/pii/S0020751915002556#s0010). Therefore, mass spectrometry provides strong support of the quality of the materials. Regarding the size of the EVs, the x-axis might lead to a confusion since it has not the same scale in both panels (A and B). We have modified panel B for a better comparison (the x-axis in both panels now ranges from 50-450 nm). This way it is better visualised that 15k EVs are bigger than 120k EVs. Indeed, the size of the particles in the gradient-separated fractions ranged from 135 nm ± 19.3 to 342 nm ± 113.9 and, particles from fractions 5-9 (the ones used for the study), range from 181 nm ± 46.4 to 232 nm ± 41.2, compared to the size of particles in the 15k pellet vesicle fraction was 274 nm ± 40.7. Hopefully we are presenting the data in a more clear way.

  1. The authors described that TSP family domains, ferritin-like domains, and 14-3-3 protein domains were common to both types of vesicles. However, the ferritin-like domain was not listed in Fig. 2B.

We would like to thank the reviewer for noticing this. It is indeed a mistake and ferritin has been removed from this sentence, which now reads “From these, TSP family domains and 14-3-3 protein domains were common to both type of vesicles”. Lines 367-368.

  1. As the authors described, tetraspanin family is common to both EV fractions, but their abundance is not the highest among the listed proteins. The rationale of why tetraspanins were chosen as potential vaccine candidates should be described.

Since we did not perform a quantitative proteomic analysis comparing both samples (120k and 15k EVs), the best approach to analyse the abundance of proteins within each sample is to look at the spectrum counting. In both cases (120k and 15k EVs), Sh-TSP-2 (a member of the CD-63 tetraspanin family) is, by far, the most abundant protein within each sample, which pointed us to study tetraspanins as vaccine candidates. Since it is mostly CD-63 tetraspanins that have been studied in vaccination studies against schistosomiasis (https://www.nature.com/articles/nm1430?proof=true), and other Platyhelminth infections such as opisthorchiasis (https://www.nature.com/articles/s41598-017-13527-5) and Echinococcus (https://pubmed.ncbi.nlm.nih.gov/22479658/), we decided not to investigate other members of the tetraspanin family such as the uroplakin tetraspanins. Furthermore, our model of schistosomiasis is a heterologous model, so we chose proteins with the highest percentage identity with S. mansoni tetraspanins. Table 1 below summarises the percentage identity of Schistosoma haematobium tetraspanin open reading frames (ORF) and large extracellular loop (LEL) with their respective Schistosoma mansoni homolog (highlighted in yellow the proteins finally chose in our study).

Table 1.

Sh-TSPs

% Identity with S. mansoni TSPs

Family

ORF

LEL

MS3_05226

86

84

Uroplakin

MS3_01370

90

81

CD-63

MS3_05289

83

81

CD-63

MS3_01153

86

82

Uroplakin

Sh_TSP-2

84

69

CD-63

MS3_09198

93

84

CD-63

  1. Why is the animal study in Fig. 4 shown as two separate trials? The rationale for this duplication should be explained.

It is standard practice when performing vaccine efficacy trials using schistosomiasis challenge experiments to do two independent trials to confirm results due to the variability inherent in these (and a lot of other) parasite challenge models (viability of challenge organisms, fitness of hosts, etc.). Likewise, it is accepted, and open and honest, practice to represent data as separate trials so as not to “disguise” any variability in individual experiments by averaging the data across trials (eg: Tran et al., Nature Med, 2006; Pearson et al., PLoS NTD, 2012; Tedla et al., Vaccines, 2020). Indeed, presenting the data as individual trials serves to highlight the reproducibility of our results and confirmation of vaccine efficacy in this study.

Moreover, this result could be an important finding but the underlying mechanism of this protection was not studied at all in this study. Studying the protection mechanism by injecting EV-derived tetraspanins would add significance to the study - for instance, immune cell activation by tetraspanins in mice.

We and others have reported extensively on the capacity of parasite EVs to mediate pathogenesis through their uptake into host cells and the ability of anti-TSP antibodies to inhibit the uptake of parasite EVs into these cells (eg: Chaiyadet et al., J Infect Dis, 2015; Eichenberger et al., Frontiers Immunol, 2018; Kifle et al., Int J Para, 2020; Kifle et al., Mol Biochem Para., 2020; Sotillo et al., Int J Para, 2020; Phumrattanaprapin et al., J Infect Dis, 2020) providing a plausible protection mechanism. We have emphasised these points more extensively in the discussion (line 519-523).

Reviewer 2 Report

The present manuscript shows novelty and is of interest in its area of vaccination against parasites. Their results are consistent with the conclusion. However, the discussion should be improved according to its results.

Author Response

Reviewer 2:

The present manuscript shows novelty and is of interest in its area of vaccination against parasites. Their results are consistent with the conclusion. However, the discussion should be improved according to its results.

We appreciate the reviewer supporting comment that this manuscript shows novelty and is of interest in its area of vaccination against parasites. As per the suggestion we have strengthened and improved the discussion section.

On behalf of the authors, thank you for your consideration.

Yours sincerely,

Mark Pearson, PhD

Research Fellow – James Cook University

Round 2

Reviewer 1 Report

All the previous comments and questions were appropriately addressed in the revised manuscript.